# Application of Pluronics for Enhancing Aqueous Solubility of Lipophilic Microtubule Destabilizing Compounds on the Sea Urchin Embryo Model

**DOI:** 10.3390/ijms241914695

**Published:** 2023-09-28

**Authors:** Marina N. Semenova, Nikolay S. Melik-Nubarov, Victor V. Semenov

**Affiliations:** 1N.K. Koltzov Institute of Developmental Biology RAS, 26 Vavilov Street, 119334 Moscow, Russia; ms@chemical-block.com; 2Department of Chemistry, M.V. Lomonosov Moscow State University, Leninskie Gory, 1/11B, 119991 Moscow, Russia; melik.nubarov@belozersky.msu.ru; 3N.D. Zelinsky Institute of Organic Chemistry RAS, 47 Leninsky Prospect, 119991 Moscow, Russia

**Keywords:** pluronics, solubilization, critical micelle concentration, toxicity, microtubule destabilization, sea urchin embryo

## Abstract

In screening, the dilution of DMSO stock solution of a lipophilic molecule with an assay medium often causes compound precipitation. To overcome the issue, the application of Pluronics as cosolvents was examined using a phenotypic sea urchin embryo assay that allows for the quick and facile evaluation of the antiproliferative effect together with systemic toxicity. Maximum tolerated concentration values for Pluronics L121, P123, and F127 were 1.4 μM, 8.6 μM, and 39.7 μM, respectively, and correlated directly with their hydrophilicity. Pluronics L121 and P123 suppressed cleavage and blastomeres retained the round shape, unlike hydrophilic Pluronic F127, which induced fertilization envelope creasing and embryo deformation that could be associated with the interaction of hydrophilic PEO units with mucopolysaccharides at the surface of sea urchin embryos. The toxicity of P123, but not of L121 and F127, was temperature-dependent and markedly increased at lower temperatures. CMC values obtained at different temperatures confirmed that the toxic effect of P123 was associated with both unimers and micelles, whereas F127 toxicity was related mainly to micelles. Evaluation using phenotypic sea urchin embryo assay revealed that potent microtubule destabilizers, namely albendazole, diarylisoxazole, and two chalcones, retained antimitotic activity after the dilution of their DMSO or 2-pyrrolidone stock solutions with 1.25% *w*/*v* Pluronic P123 or 5% *w*/*v* Pluronic F127. It was suggested that Pluronic P123 and Pluronic F127 could be used as cosolvents to improve the solubility of lipophilic molecules in aqueous medium.

## 1. Introduction

In screening for bioactivity, stock solutions of lipophilic compounds are usually prepared in DMSO at 10–20 mM concentration and subsequently diluted with an assay medium to obtain a desired nominal final concentration. However, due to insufficient aqueous solubility, the test molecule may precipitate, and its real final concentration could be significantly less than assumed, resulting in the incorrect determination of biological activity [1,2]. Our previous study involved the biological evaluation of small molecules using a phenotypic sea urchin embryo assay [3]. To enhance compound solubility in the assay medium, filtered seawater (FSW) stock solutions of tested molecules in DMSO at 10–20 mM concentration were further diluted 10 times or more with 96% EtOH and then added to the embryo suspension, taking into account the maximal tolerated concentration of DMSO and EtOH in the assay. However, the use of EtOH could be inappropriate in the other test systems. Therefore, we decided to explore other cosolvents, namely Pluronic aqueous solutions, to avoid possible the precipitation of a lipophilic molecule in a water-containing environment.

Pluronics, or poloxamers, are amphiphilic triblock copolymers that consist of a hydrophobic polypropylene oxide (PPO) segment flanked by hydrophilic polyethylene oxide (PEO) chains (Table 1). Pluronics with different lengths and ratios of PEO and PPO blocks are characterized by a unique hydrophilic–lipophilic balance (HLB), which is one of the most significant physical properties of the copolymers and is also crucial for their biological activity [4,5,6,7]. Pluronics are considered as relatively nontoxic polymers, they are FDA-approved and listed in the US and European Pharmacopoeia as pharmaceutical excipients that can be used as wetting, solubilizing, emulsifying, and stabilizing agents in drug-delivery formulations for oral, parenteral, and topical application [8]. Pluronics L121, P123, and F127 (Poloxamers 401, 403, and 407) were concluded by Cosmetic Ingredient Review (CIR) Expert Panel as safe to use in cosmetics [9]. Of these, Pluronic F127 (Poloxamer 407) was FDA-approved as a drug ingredient for oral and topical application [10]. It is the most popular triblock copolymer for the targeted delivery of hydrophobic drugs [11,12].

The ability of Pluronic unimers to self-aggregate into stable micelles in aqueous medium is a favorable feature for improving the solubility of poorly water-soluble lipophilic compounds [12]. This behavior depends on Pluronic concentration and composition as well as on external factors, such as temperature and the concentration of different inorganic ions. Micelles start to assemble at critical micelle concentration (CMC) and critical micelle temperature (CMT). At low polymer concentration and low temperature Pluronics dissolve in water as unimers [12]. Decreasing temperature caused significant CMC increase [16,17], whereas the addition of NaCl to aqueous Pluronic solution reduced both CMC and CMT values [18,19].

The solubilization capacity of Pluronics is closely related to the formation of micelles, and more hydrophobic polymers with lower PEO/PPO ratio were reported as more efficient solubilizing agents [20,21]. However, even at a temperature below CMT, the ability of Pluronics to enhance water solubility of lipophilic compound was demonstrated [22]. This effect was associated with the number of hydrophobic PPO groups (m) and decreased in the following order: F87 (m = 39.8) > F77 (m = 34.1) > F68 (m = 28.9) > F38 (m = 17.1) (see Table 1).

Pluronic application as cosolvents in a biological screening system inevitably requires the preliminary evaluation of their putative toxicity and the determination of maximum tolerated concentration to avoid undesirable adverse issues. Pluronic effects on multidrug-resistant (MDR) human cancer cells were reviewed by [6]. The interaction of the hydrophobic PPO chain of Pluronics with cellular membrane lipids provides Pluronic immersion into membranes with further translocation into cytoplasm, thereby affecting a number of cellular processes. Pluronic unimers, but not micelles, were reported to inhibit transmembrane efflux pumps to reduce mitochondrial respiration and ATP synthesis and to activate apoptotic mechanisms. However, these effects were demonstrated only on MDR cancer cells, whereas in non-MDR cells, Pluronics failed to produce such outcomes. The cytotoxic activity of a series of Pluronics against multidrug-resistant MCF7/DOX human breast cancer cells at 37 °C was studied [23]. Pluronics were found to inhibit cancer cell growth only in the form of micelles with EC_50_ (concentration that caused 50% cell death) values ranging from 0.04 to 4.8 mM, and their potency inversely correlated with HLB value. Specifically, hydrophilic Pluronic F127 with HLB = 22 (EC_50_ = 4.8 mM; 6% *w*/*v*) was significantly less toxic than Pluronic P123 with HLB = 8 (EC_50_ = 0.91 mM; 0.53% *w*/*v*). Interestingly, in addition to cytotoxic activity, Pluronics at lower concentrations were able to improve mitochondrial function, resulting in the increased viability of MCF7/DOX cells, and this effect was suggested to derive from the interaction of hydrophilic PEO blocks with the cell surface glycocalyx [24].

Literature data concerning Pluronic embryotoxicity and cytotoxicity against non-MDR cells are rather scarce and difficult to compare due to different assay conditions, particularly temperature, which markedly affects CMC value [16,25]. A number of Pluronics, including Pluronics P123 and F127, failed to exhibit cytotoxicity on bovine brain microvessel endothelial cells at a concentration of up to 1% [5]. The 1% Pluronics P85 and F127 with HLB values of 16 and 22, respectively, did not suppress the growth of DHB/K12/TRb rat colorectal carcinoma cells, whereas at the same concentration, the more hydrophobic Pluronic L61 (HLB = 3) completely reduced cell viability [26]. Similarly, Pluronic F127 exhibited relatively low cytotoxicity against HepG2 human liver carcinoma cells with no observed effect concentration (NOEC) and minimum effective concentration (MEC) values of 5% *w*/*w* and 10% *w*/*w*, respectively [27]. At 20 °C, 1% *w*/*v* Pluronic F127 caused the 10% growth inhibition of RTL-W1 rainbow trout liver cells, and its IC_50_ was estimated as >2% *w*/*v* [28]. The evaluation of Pluronic embryotoxicity against zebrafish embryos at 27 °C yielded IC_50_ and LC_50_ values of 2.4% and 5.2%, respectively, for Pluronic F127, whereas the more hydrophilic Pluronic F68 showed lower IC_50_ value of 1.6% [29]. On the contrary, the more hydrophilic Pluronic L64 (HLB = 15) was 10 times more active than Pluronic L31 (HLB = 5) in Chinese hamster ovary cells, rat lung epithelial cells, and rat macrophages [30]. The adverse effects of Pluronics featuring different structural entities on cellular metabolism were demonstrated in Caco-2 human colorectal adenocarcinoma cells and HMEC-1 immortalized human microvascular endothelial cell line [31]. The study revealed that Pluronic cytotoxicity was related to their interaction with cell membranes, the formation of ion transmembrane pores, and the reduction in the mitochondrial membrane potential. The effects were correlated with hydrophilic–lipophilic characteristics of polymers. The most hydrophilic Pluronic F127 (HLB = 22) showed no cytotoxicity up to a 1% concentration, Pluronic L64 (HLB = 15) exhibited a moderate effect, whereas the most hydrophobic Pluronic L61 (HLB = 3) was the most cytotoxic.

The ability of Pluronics P94 and F127 unimers and stabilized micelles to penetrate cells was monitored using fluorescent probes and radiolabeling [32]. For the unimers of both Pluronics, internalization via calveolar endocytosis followed by dispersion in cytoplasm was observed. Unimers of the more lipophilic Pluronic P94 were also translocated to the nucleus via passive diffusion. In contrast to unimers, stabilized micelles penetrated cells via clathrin-mediated endocytosis with the formation of the lysosomes and localized in lysosomes, accumulating in the perinuclear region. The authors assumed that Pluronic cytotoxicity could be related to the targeting of the nuclear DNA. However, the precise mechanism(s) of cytotoxicity and cellular target(s) for Pluronics remain unclear.

Considering the above literature data, we studied the toxicity and solubilization potential of Pluronics P85, L121, P123, and F127, featuring different molecular masses, lengths of hydrophilic PEO and hydrophobic PPO chains, and HLB values. For the first step, Pluronic embryotoxicity in relation to their CMC values in assay conditions was estimated using the sea urchin embryo model. Then, the ability of Pluronic cosolvents to improve the water solubility of known potent antimitotic microtubule destabilizing agents, namely albendazole **1** [33], diarylisoxazole **2** [3], and chalcones **3** and **4** [34,35] (Figure 1), was evaluated using the phenotypic sea urchin embryo assay [3].

## 2. Results and Discussion

### 2.1. Solubilization of Chalcones III and IV Using Pluronics P85, P123, and F127

Pluronics P85, P123, and F127 were reported as solubilizing agents for numerous lipophilic drugs [20]. These copolymers were selected for the assessment of their impact on aqueous solubility of chalcones **3** and **4** that exhibited pronounced cytotoxicity against human cancer cells due to the antimitotic microtubule destabilizing mechanism of action [34,35]. First, the aqueous solubility of chalcones **3** and **4** without Pluronics was determined as 344.36 mg/L (1 mM) and 88.66 mg/L (0.27 mM), respectively. Then, the impact of Pluronic P123 on chalcone solubility was estimated. This polymer was chosen because it was the most hydrophobic among the selected Pluronics and, therefore, was expected to be the most efficient [20,21].

As shown in Figure 2, the solubility of chalcones **3** and **4** increased by 13- and 19-fold, respectively, with the increasing of Pluronic P123 concentration up to 10%. Both curves reached a plateau that corresponded to the maximal amount of chalcone dissolved under given conditions. According to [36], a further increase in Pluronic P123 concentration under these conditions resulted in the formation of a gel phase, which, apparently, exerted a lower ability to dissolve lipophilic compounds.

For further solubilization experiments, we chose chalcone **IV**, as it is more hydrophobic and, therefore, less soluble. The solubility of chalcone **IV** in the presence of a fixed concentration of Pluronics depended linearly on the HLB value of the polymers, P123 > P85 > F127 (Figure 3). This relationship obviously reflected the different size of hydrophobic micellar phase in Pluronics with various HLB values. As shown previously, the size of the hydrophobic core of micelles (D_core_) gradually decreased from 11.5 nm (P123, HLB = 8) [37] to 9.2 nm (P85, HLB = 16) [38] and further to 8.2 nm (F127, HLB = 22) [37]. Clearly, this was the reason for the decrease in the solubilization capacity of Pluronics with higher HLB values.

### 2.2. Study of Pluronic Toxicity on the Sea Urchin Embryo Model

Before the evaluation of Pluronic application for the improvement of lipophilic compounds solubility in aqueous assay medium, NOEC values of Pluronics should be determined to avoid inherent cosolvent toxicity toward the chosen biological model. For biological experiments, we selected (i) Pluronic P123 as the most effective solubilizing agent (Figure 3) and (ii) Pluronic F127 as the lowest toxic triblock copolymer in various biological systems [23,29,31]. Since Pluronic P85 with fewer propylene units in the hydrophobic block (m = 40, Table 1) displayed a significantly lower solubilizing capacity than P123 (m = 65, Table 1), it was excluded from further study. The most hydrophobic polymer Pluronic L121 was selected for comparison. All chosen Pluronics featured hydrophobic PPO blocks of similar length (65–69 units, Table 1), which are essential for the interaction with the lipids of cell membranes (reviewed by [6]). Importantly, all three polymers are considered safe cosmetics ingredients [9].

The toxic effects of Pluronics on the sea urchin embryos were characterized by NOEC, MEC, and MLC values (Table 2). Developmental abnormalities caused by Pluronics F127, P123, and L121 on sea urchin embryos are illustrated in Figure 4 and described in detail in Appendix A.

Pluronic F127 at a concentration of 0.1–0.2% caused fertilization envelope softening and creasing, resulting in embryo deformation without any impact on embryonic development. The deformation was reversible and disappeared at prism–early pluteus stage. At a 0.4–1.5% concentration of F127, embryo deformation became more pronounced and irreversible, blastulae acquired a lentil-like shape (Figure 4, Appendix A), but nevertheless developed into normally swimming and viable but malformed plutei. At a 2.5% concentration of F127, the hatching of mid-blastulae was not yet affected; however, further development was arrested with the formation of slowly swimming, flattened gastrulae. F127 at the maximum tested concentration of 5% caused embryo death after 9.5 h of treatment, when control embryos reached hatched blastula stage. In summary, morphological effects of F127 on sea urchin embryos were concentration-related and independent of room temperature within the interval of 21–25 °C. Interestingly, F127 induced similar morphological abnormalities in zebrafish embryos. Specifically, in the presence of 5% F127, the zebrafish embryo chorion lost its spherical shape and became creased [29]. The authors assumed that chorion wrinkling might be the consequence of an osmotic loss of the chorion fluid. It seems reasonable to suggest that embryo deformations observed herein at a Pluronic F127 concentration of about 5% could also be explained by the osmotic pressure changes induced by this copolymer. As shown by Gu and Alexandridis [39], this Pluronic effectively caused an increase in osmotic pressure due to its high content of hydrophilic ethylene oxide units, which, subsequently, were able to induce embryo deformation and death at high concentrations of the polymer. The authors demonstrated that osmotic properties of Pluronics correlated directly with PEO content and, correspondingly, weakened with an increase in HLB.

Another explanation of phenotypic alterations caused by Pluronic F127 could be reached from the assumption that PEO blocks of hydrophilic Pluronics, especially those with high HLB values, targeted glycocalyx, a polysaccharide-based layer located outside the cell membrane, via the formation of H-bonds with polysaccharide macromolecules [24,40]. Specifically, Pluronic F68 (*n* = 153, HLB = 29, Table 1) was found to bind to hyaluronic acid, an anionic glycosaminoglycan, one of the main constituents of glycocalyx [40]. During sea urchin fertilization, sperm–egg fusion triggers the exocytosis of egg cortical granules, thereby releasing structural proteins, enzymes, and mucopolysaccharides into the extracellular space. Negatively charged mucopolysaccharides and protein hyalin undergo hydration and swelling, resulting in the elevation of cellular vitelline membrane and, eventually, the formation of a fertilization envelope that protects eggs and early embryos [41]. Hydrophilic Pluronic F127 with *n* = 200 (Table 1) could interact with mucopolysaccharides at the egg surface, thereby inducing the observed phenotypic alterations.

Hydrophobic Pluronic 123 (HLB = 8, Table 1) altered sea urchin embryo development in a completely different manner (Figure 4; Appendix A). It was significantly more toxic than F127, and its effect was temperature-dependent (Table 1). Specifically, a temperature decrease from 24 °C to 16 °C reduced MEC and MLC values by 4 and 20 times, respectively. Phenotypic impairments caused by P123 were also quite different (Figure 4). The fertilization envelope remained undamaged and clearly visible, and the blastomeres maintained their round shape, while cell division (cleavage) was inhibited up to complete cleavage arrest at MLC, followed by embryo mortality. At MEC, P123 caused the developmental delay and retardation of plutei growth.

Pluronic L121 was considered negligibly soluble in water, as it can be dissolved only at <0.001% concentration [42]. In cell culture medium, the cloud point of Pluronic L121 was estimated as 14 °C, and the turbidity of its solution was observed at 37 °C [43]. Similarly, L121 was insoluble in seawater, forming a white suspension at a concentration range appropriate for sea urchin embryo tests, but nevertheless induced sea urchin embryo developmental abnormalities. In general, the morphological changes of sea urchin embryos exposed to L121 were similar to those caused by P123 (Figure 4). However, L121 was markedly more toxic, and its effect was temperature-independent in the interval of 21–25 °C (Table 1 and Appendix A). NOEC of L121 was as low as 0.000625% (1.42 μM), and at 0.00125–0.0025%, the developmental delay and inhibition of plutei growth was detected. At a concentration of 0.005%, L121 significantly inhibited embryo development, finally causing the formation of viable motile undersized armless plutei. A further concentration increase from 0.01% to 2% resulted in mitotic arrest at various cleavage stages and embryo mortality after 8.5–10 h of treatment.

In summary, in the sea urchin embryo assay, the toxicity of Pluronics increased in the order F127 < P123 < L121 and correlated directly with their hydrophobicities (HLB values of 22, 8, and 1, respectively, Table 1). These data were consistent with the reported cytotoxicity of Pluronics F127 and P123 against MCF7/DOX human breast cancer cells with IC_50_ values of 4.8 mM and 0.91 mM, respectively [23]. In addition, Pluronic L121 at a markedly lower concentration (≥100 μg/mL or ≥0.023 mM) induced pronounced apoptotic cell death in a EL4 mouse thymoma cell line [43].

While phenotypic alterations after the exposure of fertilized eggs to Pluronics L121 and P123 were similar (Figure 2), the effect of P123, in contrast to F127 and L121, was temperature-dependent and markedly increased with decreasing temperature (Table 1 and Appendix A). According to the literature, a decrease in temperature caused the hydration of the hydrophobic PPO block of Pluronics, resulting in an increase in CMC values [12,16,44]. Therefore, one might propose that Pluronic P123 unimers were more toxic towards sea urchin embryos than micelles, in contrast to the reported predominant cytotoxicity of Pluronics in the form of micelles on MCF7/DOX human breast cancer cells [23]. To confirm this hypothesis, the determination of CMC values in assay conditions, namely in FSW and in a temperature range beneficial for sea urchin embryo development, was conducted.

### 2.3. Evaluation of Pluronic F127 and Pluronic P123 CMC in Seawater at Ambient Temperatures

Direct Pluronic F127 and Pluronic P123 CMC measurements via the micellar solubilization of the fluorescent probe 1,6-diphenylhexatriene (DPH) in seawater at different temperatures showed that the temperature decrease from 25 °C to 10 °C afforded a CMC increase by almost two orders of magnitude (Figure 5, Table 3). Hence, at a given copolymer concentration, the equilibrium concentration of unimers was elevated upon a decrease in the solution temperature. Therefore, the observed increase in the Pluronic P123 effect on sea urchin embryo development with decreasing temperatures could be attributed to an increase in the unimer concentration, suggesting the essential role of P123 unimers in embryotoxicity, whereas the additional effect of the copolymer micelles appeared at temperature higher than CMT. Generally, the ratio of micelles and unimers in the assay medium depends on the concentration of Pluronic. CMC value represents the concentration of unimers in equilibrium with micelles. At a given temperature and Pluronic concentration above the CMC, the concentration of unimers is constant and equal to the CMC. The effect of Pluronic P123 at 16 °C was observed at the concentration below CMC, confirming the toxicity of unimers only. At 21 °C, MEC of Pluronic P123 was equal to CMC, whereas at 23–24 °C, MEC was ~6 times higher than CMC (Table 2 and Appendix A). This was evidence that at 23–24 °C, both the unimers and micelles of Pluronic P123 were toxic.

**Figure 5 ijms-24-14695-f005:**
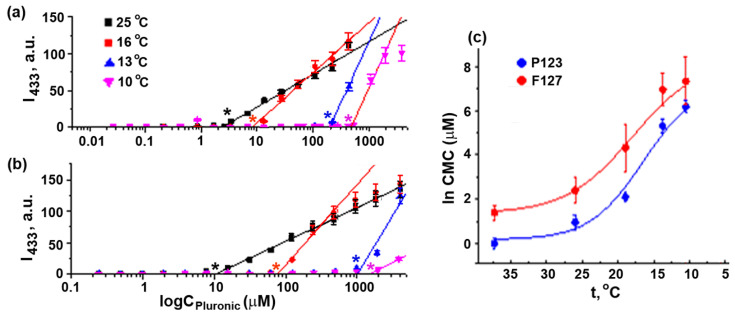
Determination of CMC values of Pluronics P123 and F127. DPH fluorescence intensity in seawater at 25 °C, 16 °C, 13 °C, and 10 °C (4) for different concentrations of Pluronic P123 (**a**) and Pluronic F127 (**b**). Asterisk: CMC. (**c**) Temperature dependence of CMC values of Pluronics P123 and F127 in seawater. CMC values at 37 °C were obtained in PBS. Vertical bars: the mean squared error (~3–5%).

The hydrophilic Pluronic F127 affected sea urchin embryo development in quite another way. It was much less toxic than Pluronic P123 and, at 21–25 °C, caused embryo abnormalities at a concentration markedly exceeding CMC (Table 2 and Appendix A), indicating the toxicity of micelles. It could be suggested that the more hydrophilic Pluronic F127 exerted its effect due to PEO affinity to polysaccharides of cellular membranes [40], particularly to mucopolysaccharides located at the surface of the sea urchin egg/embryo. As reported recently, this copolymer supported human cancer cell growth by protecting the cells from mechanical stress up to maximum tolerable concentration [24]. Therefore, Pluronic F127 micelles could interact with the surface of sea urchin eggs and blastomeres, resulting in specific morphological alterations.

In summary, Pluronics F127 and P123 toxicity data provide the identification of their tolerated concentrations (≤NOEC) that are critical to the further screening of selected lipophilic tubulin/microtubule targeting compounds on the sea urchin embryos.

### 2.4. Application of Pluronics to Enhance the Solubility of Lipophilic Tubulin-Targeting Antimitotics in the Sea Urchin Embryo Assay

Based on toxicity results, Pluronics F127 and P123 were selected for application as cosolvents to improve the aqueous solubility of lipophilic biologically active small molecules. Pluronic L121 was excluded from further examination due to its high toxicity and insolubility in FSW. For these experiments, four potent antimitotic microtubule destabilizing agents with different chemical structures and water solubility were selected, namely albendazole **1**, diarylisoxazole **2 [3]**, and chalcones **3** and **4 [34,35]** (Figure 1). The reported aqueous solubility of **1** was <1 μg/mL at pH 7.0 [45] and that of **2** was <0.2 µg/mL in PBS at 37 °C [46]. The solubility of **3** and **4** was determined as 344.36 mg/L (1 mM) and 88.66 mg/L (0.27 mM), respectively, using the procedure described in Section 2.2. The compounds were initially prepared in standard solvents, DMSO or 2-pyrrolidone, at 10 mM concentration, and then dissolved in assay medium (FSW), 96% ethanol, 5% F127, or 1.25% P123. Antimitotic effects were studied using the phenotypic sea urchin embryo assay [3]. 2-Pyrrolidone, a plant natural product and a metabolite of γ-aminobutyric acid localized in the mammalian and human brain, was chosen because of its reported application as a solubilizing excipient Kollisolv® PYR (BASF) for parenteral and oral drug formulation for animal health [47,48,49,50]. The concentrations of intermediate solutions were 1 mM, 50 μM, and 2.5 μM for the most potent compounds **1** and **2** and 1 mM and 50 μM for chalcones **3** and **4**. In all tests, the final concentrations of Pluronics in egg/embryo suspension did not exceed their NOEC values. The results are presented in Table 4. For each tested molecule, both cleavage alteration and cleavage arrest were observed at equal concentrations when 96% EtOH, 5% F127, and 1.25% P123 were used as cosolvents. In contrast, when DMSO or 2-pyrrolidone stock solutions of tested antimitotics **1**–**4** were further dissolved in FSW, the activity of all compounds significantly decreased, indicating their insufficient solubility in aqueous environment. It is noteworthy that albendazole **1I** and chalcone **4** formed a thick suspension of fine crystals in FSW at 1 mM concentration. Only 1 mM/FSW solution of chalcone **3** was clear without any visible signs of precipitation, probably due to its relatively high water solubility compared with other studied compounds. A total of 1 mM/FSW solution of diarylisoxazole **2** was opaque, followed by the precipitation of crystals in the bottom of the vial. In contrast, the 1 mM solution of **2** in 5% F127 remained clear throughout four days of storage at room temperature. The least soluble compound **1** initially dissolved in 2-pyrrolidone formed opaque 1 mM solution in 5% F127, and then fine crystals were detected after overnight storage at room temperature. Nevertheless, the quick preparation of 50 μM and 2.5 μM clear intermediate solutions of **1** in 5% F127 from 1 mM/F127 allowed us to avoid activity loss. 

## 3. Materials and Methods

### 3.1. Materials

Pluronics P85, L121, P123, and F127 were purchased from BASF (Beaumont, TX, USA). 2-Pyrrolidone and albendazole **1** were purchased from Acros Organics (Russia) at the highest commercial quality. Diarylisoxazole **2** [3] and chalcones **3** and **4** [35] were synthesized according to published procedures. 1,6-Diphenyl-1,3,5-hexatriene (DPH) and DMSO of the highest purity were purchased from Sigma-Aldrich (Darmstadt, Germany). Artificial seawater (pH 8.1) was prepared according to the published composition of Mediterranean seawater [51] and contained 0.47 M NaCl, 0.06 M MgCl_2_ × 6H_2_O, 0.01 M CaCl_2_, 0.01 M KCl, 0.04 M Na_2_SO_4_, and 0.002 M NaHCO_3_.

### 3.2. Evaluation of Aqueous Solubility of Lipophilic Microtubule Destabilizing Compounds

The absorbance of chalcone solutions in DMSO was measured using an Ultrospec 1100 Pro UV/Vis spectrophotometer. The maximum absorbance in the spectra of chalcones **3** and **4** in DMSO was observed at 367 and 350 nm, respectively (Appendix A). Molar extinction coefficients of chalcones **3** and **4** in DMSO (ε_DMSO_) were determined from the corresponding concentration–absorbance graphs of the standard 1–16 μM solutions (Appendix A) and were found to be 28,400 ± 240 M^−1^ cm^−1^ and 46,000 ± 3000 M^−1^ cm^−1^, respectively. These coefficients were used for the evaluation of chalcone solubility in the phosphate-buffered saline (PBS, 0.15 M NaCl, 10 mM Na_2_HPO_4_, pH 7.4) or in solutions of Pluronics P85, P123, and F127 in PBS. To this end, 100 mg/mL solutions of **3** and **4** in DMSO were prepared and diluted 10-fold either with neat PBS or with Pluronic solutions of various concentrations. Dilution with PBS yielded the precipitation of the most part of the solute. The precipitated chalcones were removed via centrifugation at 9000 rpm for 5 min. The supernatant represented a saturated solution of the chalcone in the aqueous medium. Then, the supernatants were 10–1000-fold diluted with DMSO, their absorbance was measured at 367 nm for **3** or 350 nm for **4**, and the aqueous solubility of chalcones was calculated using the ε_DMSO_ values obtained previously. The solubility values were measured in 2–4 repetitions and presented in the graphs as the mean ± SD for *p* < 0.05. 

### 3.3. CMC Determination Procedure

CMC values at different temperatures in seawater were determined using the DPH fluorescence assay [52]. Briefly, a 13 mM solution of DPH in acetone was diluted 1000-fold with seawater and stirred at ambient temperature in a fume hood for 1 h until acetone evaporation. A series of 1 mL aliquots of polymer solutions with gradually decreasing concentration were incubated in cooling chambers at a selected temperature for 50–60 min and then mixed with 1 mL of DPH solution. The samples were kept at the selected temperature for an additional hour and DPH fluorescence was measured at λ_ex_ = 366 and λ_em_ = 433 nm. CMC was defined as the concentration corresponding to the intersection of the horizontal and linearly ascending segments of the dependence of fluorescence intensity on Log(C(polymer)). All measurements were performed in triplicates and standard deviation was shown in the graphs. Confidence intervals for CMC values (δCMC) were obtained as the fitting errors of the linear part of the curves and were calculated from the slope (S ± δS) and intercept (I) errors (δS and δI, correspondingly) as $\delta CMC=\frac{\delta (I)\cdot S-\delta (S)\cdot I}{I^{2}}$. The calculations were carried out using Origin 7.0 software.

### 3.4. Phenotypic Sea Urchin Embryo Assay

Adult sea urchins, *Paracentrotus lividus* L. (Echinidae), were collected from the Mediterranean Sea on the Cyprus coast and kept in an aerated seawater tank. Obtaining the gametes, fertilization, and embryo rearing were performed as described previously [3]. Tests were conducted in 6-well plates containing 5 mL aliquots of embryo suspension at a concentration of ≤2000 embryos/mL in each well.

Pluronic solutions were prepared in filtered seawater (FSW) as follows: 5% *w*/*v* (3.97 mM) Pluronic F127, 1.25% and 5% *w*/*v* (2.15 mM and 8.6 mM) Pluronic P123, and 1.25% and 5% *w*/*v* (2.85 mM and 5.7 mM) Pluronic L121 were stored at room temperature (Pluronics F127 and P123) or at +4 °C (Pluronic L121). Toxic effects were assessed by exposing fertilized eggs (5−18 min after fertilization) to 2-fold decreasing concentrations of Pluronics. Embryo development was monitored at room temperature until the beginning of active feeding (four-arm mid-pluteus stage). The effects were detected at the following developmental stages: cleavage (2.5 and 5.5–6 h after fertilization), hatched mid-blastula (8.5–9.5 h), mesenchyme blastula (~11 h), prism (20–21 h), early pluteus (25–27 h), and four-arm pluteus (34–35 h) (Appendix A). The effects were estimated quantitatively as no observed effect concentration (NOEC), minimum effective concentration (MEC), and minimum lethal concentration (MLC).

Stock solutions of albendazole **1** and isoxazole **2** were prepared in DMSO and 2-pyrrolidone at 10 mM concentration, and stock solutions of chalcones **3** and **4** were prepared in DMSO at 10 mM concentration. The stock solutions were further diluted with 96% EtOH, 5% Pluronic F127, 1.25% Pluronic P123, or FSW to obtain the final concentration range of a compound in embryo suspension relevant to the assessment of the antimitotic activity. The NOEC values of DMSO, 2-pyrrolidone, and EtOH in the in vivo assay were determined to be 0.2% *v*/*v*, 0.1% *v*/*v*, and 0.5% *v*/*v*, respectively. In all tests, the final concentration of the solvents in egg/embryo suspension did not exceed NOEC. The precipitation of compounds was analyzed via both visual and microscopic examination. The antimitotic activity of compounds **1**–**4** was assessed by exposing fertilized eggs to 2-fold decreasing concentrations of the compound within the concentration range applicable for clear assessment of the effect: 0.0005–0.01 μM for compounds **1** and **2**, 0.005–0.2 μM for chalcone **3**, and 0.05–1 μM for chalcone **4**. Cleavage alteration and arrest were clearly detected at 2.5 h and 5.5 h after fertilization, when control embryos reached 8-cell and early blastula stages, respectively. The effects were estimated quantitatively as the minimum (threshold) effective concentration (MEC), resulting in cleavage alteration or full mitotic arrest. At these concentrations, all tested microtubule destabilizers caused 100% cleavage alteration/arrest and embryo death before hatching, whereas at 2-fold lower concentrations, the compounds failed to produce any effect. The embryos were observed with a Biolam light microscope (LOMO, St. Petersburg, Russia). Microphotographs were obtained using an AmScope binocular microscope with an MU500 digital camera (United Scopes LLC, Irvine, CA, USA). Sea urchin embryo assay data are available at http://www.zelinsky.ru. Experiments with the sea urchin embryos fulfill the requirements of biological ethics. The artificial spawning does not cause animal death, embryos develop outside the female organism, and both post-spawned adult sea urchins and the excess of intact embryos are returned to the sea, their natural habitat.

## 4. Conclusions

A possible application of Pluronics as cosolvents to enhance the solubility of lipophilic biologically active compounds in aqueous assay medium was studied using the sea urchin embryo model. It was shown that Pluronics P85, P123, and F127 improved the solubilization of chalcones **3** and **4** in water-containing solutions, and the solubility increase depended on the HLB value. The most hydrophobic Pluronic, P123, was the most effective. Particularly, in the presence of 5% Pluronic 123, the solubility of chalcones **3** and **4** in aqueous medium was enhanced by an order of magnitude. To exclude undesirable adverse effects, the toxicity of Pluronics toward sea urchin embryos was evaluated and NOEC values were determined. The maximum tolerated concentration of Pluronics correlated directly with their hydrophobicity, varying from 1.4 μM for L121 to 8.6 μM for P123 and 39.7 μM for F127.

More hydrophobic Pluronics L121 and P123 impaired sea urchin embryo development as follows: cell division (cleavage) was inhibited up to complete cleavage arrest at MLC, followed by embryo mortality, whereas the fertilization envelope remained undamaged and clearly visible and blastomeres maintained their round shape. In contrast, the most hydrophilic and the less toxic Pluronic F127 caused fertilization envelope softening and creasing, resulting in embryo deformation that was reversible at lower concentrations. This effect could be attributed to the interaction of hydrophilic PEO units of F127 with mucopolysaccharides located at the surface of sea urchin eggs and embryos. The toxic effect of P123, but not of F127, was temperature-dependent and increased substantially with temperature decrease. CMC values in seawater at different temperatures suggested that both the unimers and micelles of P123 were responsible for the developmental alterations, whereas F127 toxicity was attributed mainly to micelles. The evaluation of antimitotic microtubule destabilizing the activity of potent tubulin-targeting agents, namely albendazole **1**, diarylisoxazole **2**, and chalcones **3** and **4**, on sea urchin embryos using both DMSO and 2-pyrrolidone stock solutions and different intermediate solvents provided evidence that 5% *w*/*v* Pluronic F127 and 1.25% *w*/*v* Pluronic P123 can be considered appropriate cosolvents for the solubilization of lipophilic molecules for screening in a water-containing assay medium. The application of Pluronic F127 seems to be more convenient due to its low toxicity and powder state.

## Figures and Tables

**Figure 1 ijms-24-14695-f001:**
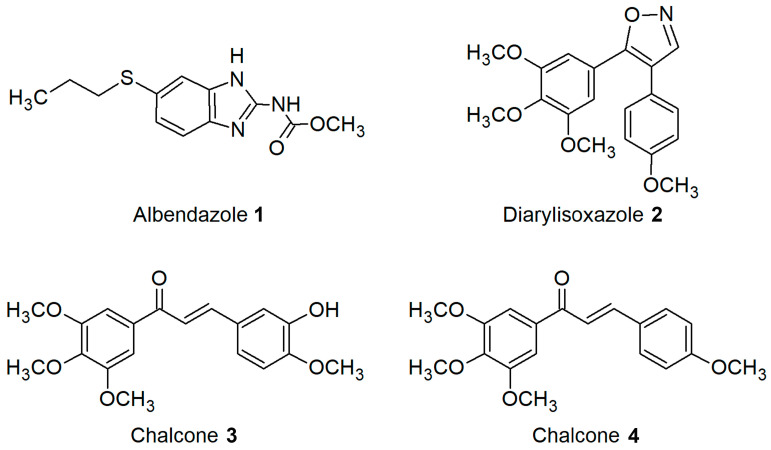
Structures of studied antimitotic anti-tubulin agents.

**Figure 2 ijms-24-14695-f002:**
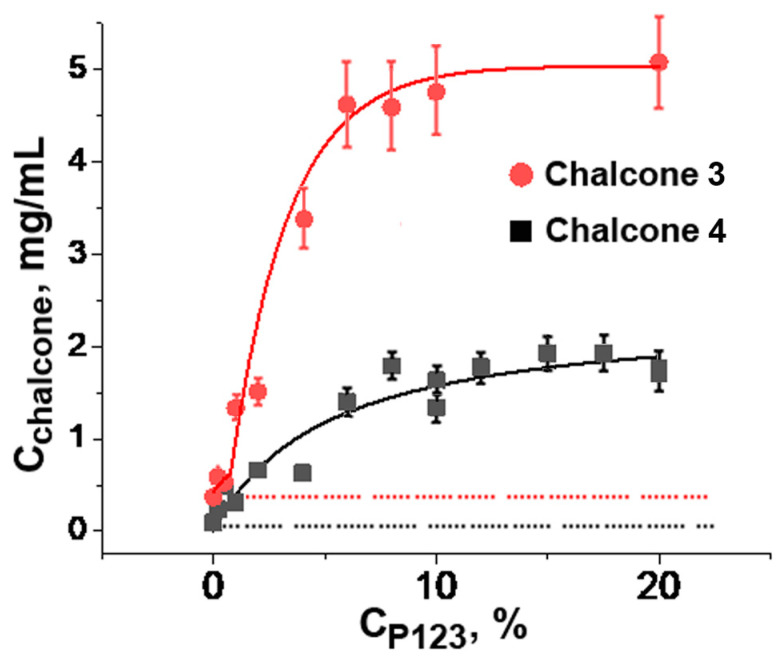
Solubility of chalcones **3** and **4** at different concentrations of Pluronic P123 in phosphate-buffered saline (10 mM Na_2_HPO_4_, 150 mM NaCl, pH 7.4) at 25 °C. C_chalcone_: Concentration of the saturated chalcone solution in Pluronic 123 after removal of insoluble precipitate from the initial 10 mg/mL chalcone suspension. Vertical bars: the mean squared error.

**Figure 3 ijms-24-14695-f003:**
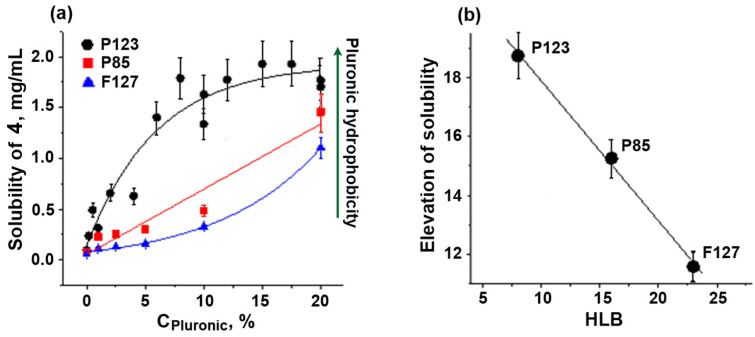
Solubility of chalcone **4** at different concentrations of Pluronics P85, P123, and F127 (**a**) and the correlation of solubility elevation with HLB values (**b**). Vertical bars: the mean squared error.

**Figure 4 ijms-24-14695-f004:**
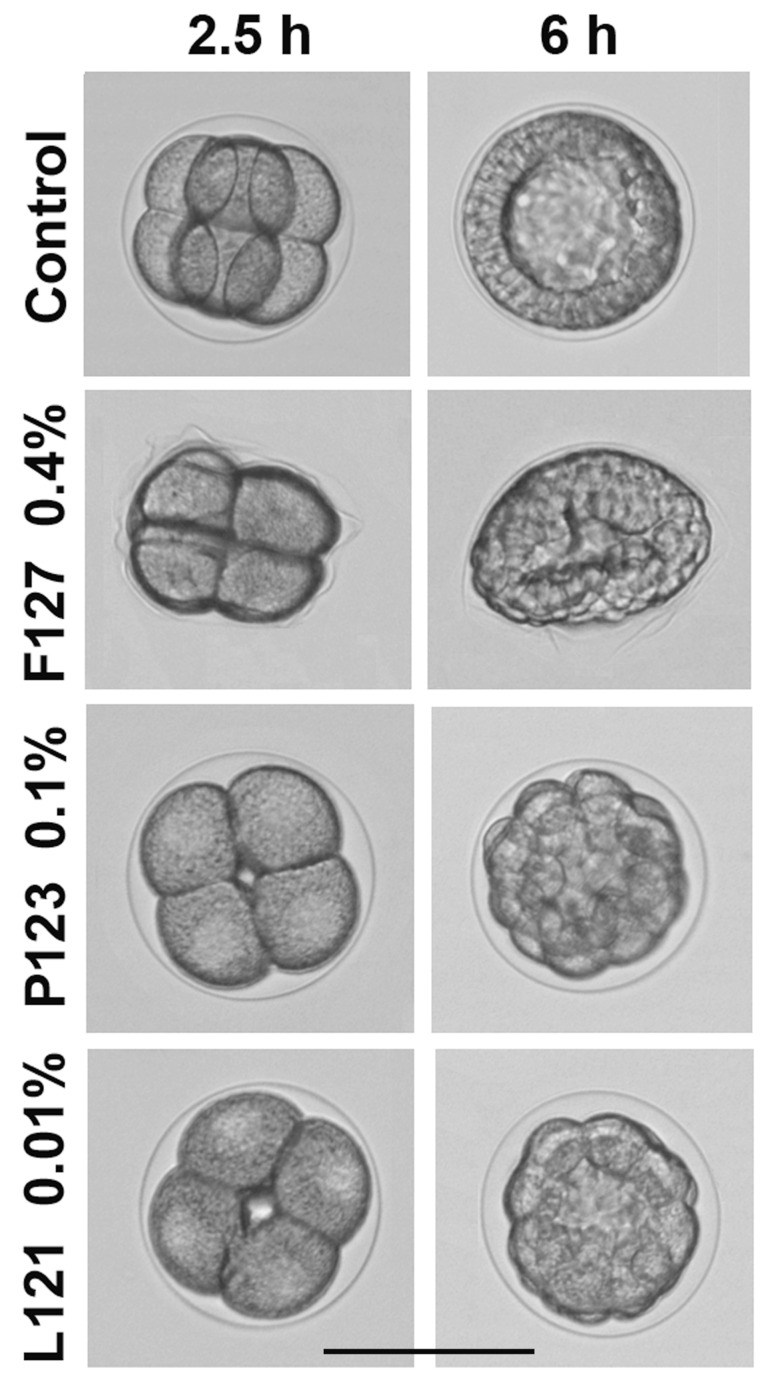
Representative morphological alterations of sea urchin embryos treated with Pluronics F127, P123, and L121. Fertilization envelope softening and embryo deformation caused by F127 at MEC = 0.4% *w*/*v* (316 μM). Developmental arrest and formation of aberrant blastulae with clearly visible fertilization envelope after exposure to P123 and L121 at MLC = 0.1% *w*/*v* (172.5 μM) and 0.01% *w*/*v* (22.7 μM), respectively. Fertilized eggs treatment. Incubation temperature: 23–24 °C. Average embryo diameter: 115 μm. Scale bar: 100 μm.

**Table 1 ijms-24-14695-t001:** Characteristics of selected Pluronics ^a^.

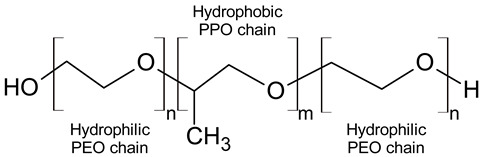					
**Pluronic**	**Poloxamer**	**MW**	**Average m ^b^**	**Average *n* ^c^**	**HLB ^d^**
*Pluronics studied in this work*					
P85	P235	4600	39.7	52.3	16
L121	P401	4400	68.3	10	1
P123	P403	5750	69.4	39.2	8
F127	P407	12600	65.2	200.4	22
*Pluronics from the literature cited in this work*					
L31	P101	1100	17	2.5	5
F38	P108	4700	17.1	84.3	31
L61	P181	2000	31	4.5	3
L64	P184	2900	30	26.4	15
F68	P188	8400	28.9	152.7	29
F77	P217	6600	34.1	104.9	25
F87	P237	7700	39.8	122.5	24
P94	P284	4600	47	42	14
P103	P333	4950	59.7	33.8	9
P105	P335	6500	56	73.9	15

^a^ Data from [8,9,13,14,15]. ^b^ Average number of hydrophobic propylene oxide units. ^c^ Average number of hydrophilic ethylene oxide units. ^d^ HLB: hydrophilic–lipophilic balance.

**Table 2 ijms-24-14695-t002:** Effects of Pluronics F127, P123, and L121 on the sea urchin embryos ^a^.

Pluronic	Mean Temperature, °C	NOEC		MEC		MLC		CMC, μM ^b^	
		% *w*/*v*	μM	% *w*/*v*	μM	% *w*/*v*	μM	% *w*/*v*	μM
F127	23	0.05	39.7	0.1; 0.4 ^c^	79; 316 ^c^	5	3970	0.023	18.7 ± 3
P123	24	0.005	8.6	0.01–0.02	17.25–34.5	0.1	172.5	0.0017	3.0 ± 0.5
	23	0.005	8.6	0.01–0.02	17.25–34.5	0.05	86	0.0017	3.0 ± 0.5
	21	0.00125	2.15	0.0025–0.005	4.3–8.6	0.01–0.02	17.25–34.5	0.0025	4.3 ± 1
	16	0.00125	2.15	0.0025	4.3	0.005	8.6	0.023	40 ± 10
L121 ^d^	23	0.000625	1.42	0.00125	2.84	0.01	22.7	<0.0004	<1

^a^ The sea urchin embryo assay was conducted as described previously [3]; fertilized eggs were exposed to two-fold decreasing concentrations of Pluronics; duplicate measurements showed no differences in NOEC, MEC, and MLC values. ^b^ CMC values for Pluronics F127 and P123 in artificial seawater were calculated from the graphs presented in Figure 5c (see below, Section 2.3). Solubility limit is presented as insoluble in seawater Pluronic L121. ^c^ At 0.1–0.2% (79–158 μM) the effect of Pluronic F127 (embryo deformation) was reversed, no difference between treated and intact embryos/larvae were observed from prism stage to early pluteus stage. ^d^ Approximate concentration values are presented, since Pluronic L121 was insoluble in FSW, forming white milky-like suspension.

**Table 3 ijms-24-14695-t003:** CMC values of Pluronics P123 and F127 in artificial seawater at different temperatures.

Pluronic	CMC/Seawater							
	25 °C		18 °C		13 °C		10 °C	
	% *w*/*v*	μM	% *w*/*v*	μM	% *w*/*v*	μM	% *w*/*v*	μM
F127	0.0139 ± 0.001	11 ± 1.3	0.095 ± 0.01	75 ± 10	1.3 ± 0.1	1025 ± 120	1.90 ± 0.17	1520 ± 130
P123	0.0011 ± 0.0001	1.9 ± 0.2	0.018 ± 0.002	31.1 ± 3	0.063 ± 0.1	108.9 ± 12	0.273 ± 0.03	412 ± 50

**Table 4 ijms-24-14695-t004:** Effects of compounds **I**–**IV** dissolved in different cosolvents on sea urchin embryos.

Cmpd	Solvent ^a^	Cosolvent	MEC, μM	
			Cleavage Alteration	Cleavage Arrest
**1**	DMSO	FSW	>0.01	>0.01
		96% EtOH	0.002	0.005
		5% F127	0.002	0.005
		1.25% P123	0.002	0.005
	2-Pyrrolidone	FSW	0.005	>0.01
		96% EtOH	0.002	0.005
		5% F127	0.002	0.005
**2**	DMSO	FSW	0.005	0.01
		96% EtOH	0.001	0.005
		5% F127	0.001	0.005
		1.25% P123	0.001	0.005
	2-Pyrrolidone	FSW	0.01	>0.01
		96% EtOH	0.001	0.005
		5% F127	0.001	0.005
**3**	DMSO	FSW	0.02	0.2
		96% EtOH	0.01	0.05
		5% F127	0.01	0.05
		1.25% P123	0.01	0.05
**4**	DMSO	FSW ^b^	>0.2	Not available
		96% EtOH	0.1	1
		5% F127	0.1	1
		1.25% P123	0.1	1

^a^ Stock solutions were prepared in DMSO or 2-pyrrolidone at a 10 mM concentration. Compound concentrations in intermediate solutions were 1 mM, 50 μM, and 2.5 μM for albendazole **1** and diarylisoxazole **2** and 1 mM and 50 μM for chalcones **3** and **4**. ^b^ It was impossible to test compound **4** at a concentration of >0.2 μM due to its insufficient solubility in FSW (pronounced precipitation in 1 mM/FSW intermediate solution was observed).

## Data Availability

Data is contained within the article or Appendix A.

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
