# Peer review of "Application of Pluronics for Enhancing Aqueous Solubility of Lipophilic Microtubule Destabilizing Compounds on the Sea Urchin Embryo Model"

_ijms, 2023, doi:10.3390/ijms241914695_

Round 1

Reviewer 1 Report

CMC os pluronics changes the bioavailability coumpunds with low solubility which is used together with DMSO.

The authors present a very interesting work on increasing the solubility of antimitotic antitubulin agents with mixtures of DMSO and pluronics. The idea is that DMSO by itself isn't well tolerated and the pluoronic can enhance the solubility of the agents, without consequent toxic effects of the DMSO. For that reason, the authors determined the NOEC concentrations of the plutonics. The authors also correctly determined the importance of the CMC and correlated that with the hydrophobicity etc.

However for the crucial data that checks the effect of the micelle of the pluronic single data points, without any statistical analysis is presented, rather than graphs, with errors. This leaves me in doubt about the veracity of the data. 

Reviewer 2 Report

The authors investigate the application of Pluronics for enhancing aqueous solubility of lipophilic biologically active compounds. They found that Pluronics P85, P123, and F127 improved solubilization of antitubulin agents such as chalcones III and IV in water containing solutions. The toxicity of Pluronics toward sea urchin, Paracentrotus lividus L. (Echinidae), embryos was evaluated. The maximum tolerated concentration of Pluronics was also determined.

I have a few recommendations for the improvement of the paper:

1.       page 2, 2nd paragraph: “It is the most popular triblock copolymer for targeted delivery of hydrophobic drugs [11,12].” The introduction will benefit mentioning other common polymers for drug delivery such as poly ethylene glycol ( PEG). Are there special reasons to avoid PEGs in this study?

2.       The authors could find helpful a recent study on enhancing the aqueous solubility of lipophilic biologically active compounds from the perspective of green formulations: Composition-Switchable Liquid Crystalline Nanostructures as Green Formulations of Curcumin and Fish Oil, ACS Sustainable Chem. Eng. 2021, 9, 44, 14821–14835, https://doi.org/10.1021/acssuschemeng.1c04706

Abstract: Please rewrite the first two sentences “In screening, dilution of DMSO stock solution of a lipophilic molecule with an assay medium often causes compound precipitation. To overcome the issue, the application of Pluronics as cosolvents was examined on a sea urchin embryo model.” for a better readability. Now they sound fragmented and raise the question why sea urchins were needed for this study.

Round 2

Reviewer 1 Report

No comment